# Publisher Transparency among Communications and Library and Information Science Journals: Analysis and Recommendations

**Alexandre López-Borrull** [1,*] , **Mari Vállez** [2] , **Candela Ollé** [1] and **Mario Pérez-Montoro** [2]

1 Faculty of Information and Communication Sciences, Universitat Oberta de Catalunya, 08035 Barcelona, Spain; collec@uoc.edu
2 Faculty of Information and Audiovisual Media, Universitat de Barcelona, 08014 Barcelona, Spain; marivallez@ub.edu (M.V.); perez-montoro@ub.edu (M.P.-M.)
\* Correspondence: alopezbo@uoc.edu

**Abstract:** The principal goal of the research study is to analyze the transparency of a selection of academic journals based on an analysis model with 20 indicators grouped into 6 parameters. Given the evident interest in and commitment to transparency among quality academic journals and researchers' difficulties in choosing journals that meet a set of criteria, we present indicators that may help researchers choose journals while also helping journals to consider what information from the editorial process to publish, or not, on their websites to attract authors in the highly competitive environment of today's scholarly communication. To test the validity of the indicators, we analyze a small sample: the Spanish Communications and Library and Information Science journals listed in the Scimago Journal Rank. The results confirm that our analysis model is valid and can be extrapolated to other disciplines and journals.

**Keywords:** academic journals; transparency; Plan S





## 1. Introduction

In recent years, the scholarly communication ecosystem has undergone a series of changes, often exacerbating aspects that have been in crisis since the late 20th century [1]. Open access (defined as a movement that fights to put the maximum number of scientific articles and content open for free [2]) or paradigms such as open science could change this ecosystem even more and favor the emergence of new actors. The European Commission defines open science as 'a new approach to the scientific process based on cooperative work and new ways of disseminating knowledge, improving accessibility to and re-usability of research outputs by using digital technologies and new collaborative tools' [3]. In such a fast-changing environment, choosing a journal in which to publish becomes more complex. Likewise, the use of the impact factor for comparison and differentiation between journals has been called into question [4]. In fact, in recent years, more than ever, initiatives have been promoted that are critical of the use of the impact factor for the evaluation of research and try to create and promote alternatives. These alternatives evaluate research based on the use of the article, rather than journal, or take a qualitative, rather than a quantitative, approach. For example, DORA [5] and the Leiden Manifesto [6] have opened up discussions at an international level. In some cases, this debate has also been conducted at a national level [4]. So, there is the opportunity to analyze the quality of journals in other terms. For example, there is Plan S, which promotes a system for quality open access academic journals [7].

### 1.1. Transparency as a Vector and Value in the Evolution of Scholarly Communication

For academic journals, the arrival of the internet meant both disruption and an opportunity to optimize the process of scholarly communication, improving the dissemination of

knowledge through online publication [8,9]. This also led to the opportunity to promote open access to academic publications and changes in both scientific policies and business models [10]. The advances in technology have generated and consolidated open-science initiatives [11], including debate on the peer-review model [12]. This is not only an advance in terms of scientific output (whether articles or research data) but also a clear improvement in quality and transparency. It forms part of a process that affects transparency, open data (the process of defining how scientific data may be published and re-used without price or permission barriers [13]), and open government (high levels of transparency and mechanisms for public scrutiny and oversight [14]).

Academic journals also have a wide range of (often invisible) internal data, such as their budgets, number of articles rejected, number of reviewers, response time, etc. Some of these data are perceived as for internal use only or important to maintain a competitive edge in the process. These are just a few examples of data that are available to journals' editorial boards but which are not shared with readers or prospective authors. Scholarly communication (and journals in particular) has a set of ethical challenges that can be linked to attitudes but also to the culture of sharing data and information [15]. Now that transparency has become a key element of management, especially in the public sphere, a positive move would see academic journals offer as much information as possible about the different processes involved in their publication; this would raise journals' prestige and make it easier to assess their quality [16]. As Fosang and Colbran point out [17], transparency is the key to quality.

The high number of journals belonging to publishers of public universities must also be considered since the debate on the governance and financing of universities directly affects journals. Formerly, they were very vocational with a low budget, but they are now increasingly competitive, professional journals with a more global vision. If journals are paid for with public funds, then this must also be taken into account in terms of transparency, in the same way that national laws may require universities to have transparency portals (and open data). In recent years, the academic journal ecosystem has gone through a series of changes that have made the process of choosing a journal for publishing research results more complex [1]. Changes in the ecosystem of journals, greater pressure to publish in journals in higher quartiles, the push for open access publishing, and a lack of knowledge regarding journals are some ideas that appear in various studies that deal with the decisions doctoral students make when publishing their articles [18,19]. This affects Ph.D. students in particular, especially those opting to publish their thesis as a compendium of publications [20]. Threats to scholarly communication, including predatory journals and publishers, also need to be considered. Having more and better information on the editorial processes would help improve the decision-making process when authors look to choose a journal to publish in. Different assessment models are now being promoted, which are critical of the weight of the impact factor and dependence on quartiles; they offer new ways to evaluate journals and choose where to publish [5,6]. Authors faced with similar rankings in indexes and undifferentiated metrics may end up making their choice for other reasons. Undoubtedly, the speed of publication and the quality of the peer review are crucial, especially at a time when some assessment agencies seem to be changing their criteria in response to the emergence and consolidation of new publishers and business models. The new transformative agreements between countries, universities, and publishers to establish payment quotas for a greater number of open articles is a good example of change. Transformative agreements (also known as 'offsetting', 'read and publish', or 'publish and read' agreements) have shifted the focus of scholarly journal licensing from cost containment towards open access publication [21]. Undoubtedly, the pressure to publish with open access has led publishers to a change. It is to be expected that open science and its implementation will accelerate this transformation. All of this is also happening while authors are expressing concern about the possibility of their work falling into the hands of predatory journals [22].

The quality indicators used in assessing academic journals usually include transparency, but a wider vision of transparency is required, especially in terms of the use of public funding. For example, one of the quality indicators for Plan S-compliant open access journals involves information transparency. Plan S is an initiative of Coalition S, the European Research Council (ERC), and several European state agencies. It has attained a prominent place in the field of research and scholarly communication since its first version was published on 4 September 2018. The proposal aims to accelerate the transition to open access and ensure that, from 2021, all scientific publications derived from publicly funded projects are published in open access immediately. Plan S allows the publication of the article in three ways, one of which is publication in quality open access journals. It lists several aspects to be met by journals; some are mandatory, and others are strongly recommended.

In fact, Plan S itself presents as one of its principles the control of spending on scientific publications, declaring that publication rates should be standardized and limited. As a result, there has been some movement among publishers towards greater transparency regarding prices and margins. For example, MDPI provides a breakdown of the cost of the publication of its articles [23]

### 1.2. Transparency as a Metric for Analyzing and Comparing Journals

We propose the creation of a series of indicators to assess the transparency of a journal's editorial process. The indicators are not linked to the content of the articles or the supplementary data but to the information the journal itself supplies on the process. Some of these elements are closely linked to elements compiled by databases and quality agencies. However, as far as we know, at present, no classification of this type exists, despite the research on transparency in academic journals that has been published [24–26]. It could help to improve and optimize journals and be used in the form of a checklist to improve the information they provide.

As a preliminary measure to test its effectiveness and ability to differentiate among similar journals, we propose analyzing Social Sciences journals indexed in Scimago Journal Rank, and specifically Communication, and Library and Information Sciences (LIS) journals, in order to see whether these journals are opting to offer the data generated when processing and publishing articles. This will enable us to gather a sample to show how far this practice is being implemented in journals. Based on this analysis, we will present proposed indicators that journals could use to increase transparency in their processes. Our goals are:

-   To develop a proposal to improve journals, enabling them to have a transparency policy.
-   To create a closed set of indicators for studying and comparing the transparency of academic journals.
-   To study the level of transparency of Spanish Communication and LIS journals indexed in the Scimago Journal Rank.

## 2. Methodology

The methodology used for this study centered on the analysis of informational content in the website pages of the selected journals. The selection process focused on journals in the fields of Communication and Library and Information Sciences published in Spain and used Scimago Journal and Country Rank. Duplicate journals across both disciplines were eliminated. The final corpus analyzed had 25 journals, which were examined by 4 assessors in April 2021.

The assessment used 6 parameters (own and external human resources, financial resources, efficiency of the editorial process, quality of the editorial process, transparency of policies, and transparency of article metadata) and a total of 20 indicators, given the values 0/1. Descriptions of the indicators can be found in Table 1.

**Table 1.** List and description of the indicators used for analysis of the journals.

| Indicator | Title | Description of Requirement |
|:---:|:---:|:---:|
| Ind1 | Editorial board | The members and membership of the journal's editorial board are available. |
| Ind2 | Reviewers | The names of the reviewers are available. |
| Ind3 | Information on the reviewers | The affiliation and/or origin of the reviewers are available. |
| Ind4 | Article publication charge (APC) | The article publication charges (APCs) are available. |
| Ind5 | Itemizing costs of the publication | The costs associated with article processing and publication are available (according, for instance, to FOOA [27]). |
| Ind6 | Funding of the publication | The publication's funding sources (public, private, etc.) are available. |
| Ind7 | Response time | The estimated response time for the decision to publish articles is available. |
| Ind8 | Rejected articles | The number or percentage of articles rejected by the journal is available. |
| Ind9 | Collection of annual data on the publication | An annual information/data/stats report or infographic from the journal is available. |
| Ind10 | Manuscript review and selection process | The criteria applied during the manuscript review and selection process are available. |
| Ind11 | Sections of the publication | The characteristics that the manuscripts must meet to be published in the different sections of the journal are available. |
| Ind12 | Plagiarism | There are mechanisms to detect plagiarism. |
| Ind13 | Indexing | Detailed information on the journal's indexing is available. |
| Ind14 | Code of ethics | The publication's code of ethics is available. |
| Ind15 | License type | The type of transfer of authors' rights is made explicit. |
| Ind16 | Open access policies | The publication's open access policy is made explicit. |
| Ind17 | Open-data policies | The publication's open-data policy is made explicit. |
| Ind18 | Co-authorship | Each author's role in articles must be reported. |
| Ind19 | Monitoring self-citation | The journal has a self-citation policy. |
| Ind20 | Article metrics | Article metrics are reported. |

Each journal was assessed twice, by two different assessors, to ensure the same criteria were applied, and in cases where they did not agree, a third assessor re-assessed the indicator in question.

The indicators used were based on previous work by López-Borrull et al. [28] and on the review, analysis, and subsequent selection of the transparency indicators of the Directory of Open Access Journals (DOAJ), the Spanish Foundation for Science and Technology (FECYT, from its Spanish initials), Web of Science (WOS), SCOPUS, and Plan S (see Table 2). FECYT is a public foundation, coming under the umbrella of the Ministry of Science and Innovation. Its mission is to promote scientific research of excellence. FECYT organizes the Call for the Evaluation of Editorial and Scientific Quality to obtain the FECYT Quality Seal. This seal means that academic journals comply with a series of indicators that FECYT

defines and renews periodically. It should be noted that we could have added other sources, such as Latindex, but we considered that Table 2 was already comprehensive enough.

**Table 2.** Indicators used for analysis of journals and correspondence with sources of information.

| Indicator | Parameter | Title | DOAJ | FECYT | WOS | SCOPUS | Plan S |
|---|---|---|---|---|---|---|---|
| Ind1 | Own and external human resources | Editorial board | X | X | X | X | X |
| Ind2 | | Reviewers | - | X | - | - | - |
| Ind3 | | Information on the reviewers | - | - | - | - | - |
| Ind4 | Financial resources | Article publication charge (APC) | X | X | - | - | X |
| Ind5 | | Itemizing costs of the publication | - | - | - | - | X |
| Ind6 | | Funding of the publication | - | - | - | - | X |
| Ind7 | Efficiency of the editorial process | Response time | - | - | - | - | X |
| Ind8 | | Rejected articles | - | - | - | - | X |
| Ind9 | | Collection of annual data on the publication | - | - | - | - | X |
| Ind10 | Quality of the editorial process | Manuscript review and selection process | X | X | X | X | X |
| Ind11 | | Sections of the publication | - | X | - | X | - |
| Ind12 | | Plagiarism | X | X | X | X | X |
| Ind13 | | Indexing | - | - | - | - | - |
| Ind14 | Editorial policy | Code of ethics | X | X | X | X | X |
| Ind15 | | License type | X | X | - | - | X |
| Ind16 | | Open access policies | X | X | - | - | X |
| Ind17 | | Open-data policies | - | - | - | - | X |
| Ind18 | Metadata | Co-authorship | - | X | - | - | - |
| Ind19 | | Monitoring self-citation | X | - | X | X | - |
| Ind20 | | Article metrics | - | - | - | - | - |

## 3. Results and Discussion

The methodology used enabled us to obtain interesting results. Specifically, these results related to the transparent assessment of the proposed indicators, to the generation of a general transparency index for the best and worst journals in the selected corpus, to a distribution analysis of the journals, and to a study of the correlation between the level of transparency and other external quantitative indicators applicable to the journals in the corpus. Our results show that the analysis system developed for this study is effective in assessing the transparency of the proposed indicators. We can analyze the level of compliance with the 20 indicators and identify in the analyzed corpus of journals which indicators have the worst scores in this analysis, which have the best, and which indicators obtained intermediate results.

In relation to this, we also propose a visual representation of this analysis of compliance with the indicators through the creation of a bar chart showing the value obtained for each indicator and grouping each parameter's indicators by color (as shown in Figure 1). This enables a nominal comparison between the indicator values and the distribution of results by parameter. In our study, we can generate the corresponding graph (Figure 1)

and see the levels of compliance with or implementation of the 20 indicators in the selected corpus of journals. The figure is designed to show which indicators are already well established and which journals need to devote more effort to this end. With this analysis and proposed visualization, we can see that the indicators with the lowest compliance are 2 and 3 (reviewers), 5 (itemizing costs), 17 (open-data policies), and 19 (monitoring self-citation). The indicators with an intermediate score are 4 (APC), 12 (plagiarism), 7 (response time), and 20 (metrics). The indicators with the highest compliance are 1 (editorial board), 10 (article review and selection process), 13 (indexing), 14 (code of ethics), 15 (license type), and 16 (open access policies).

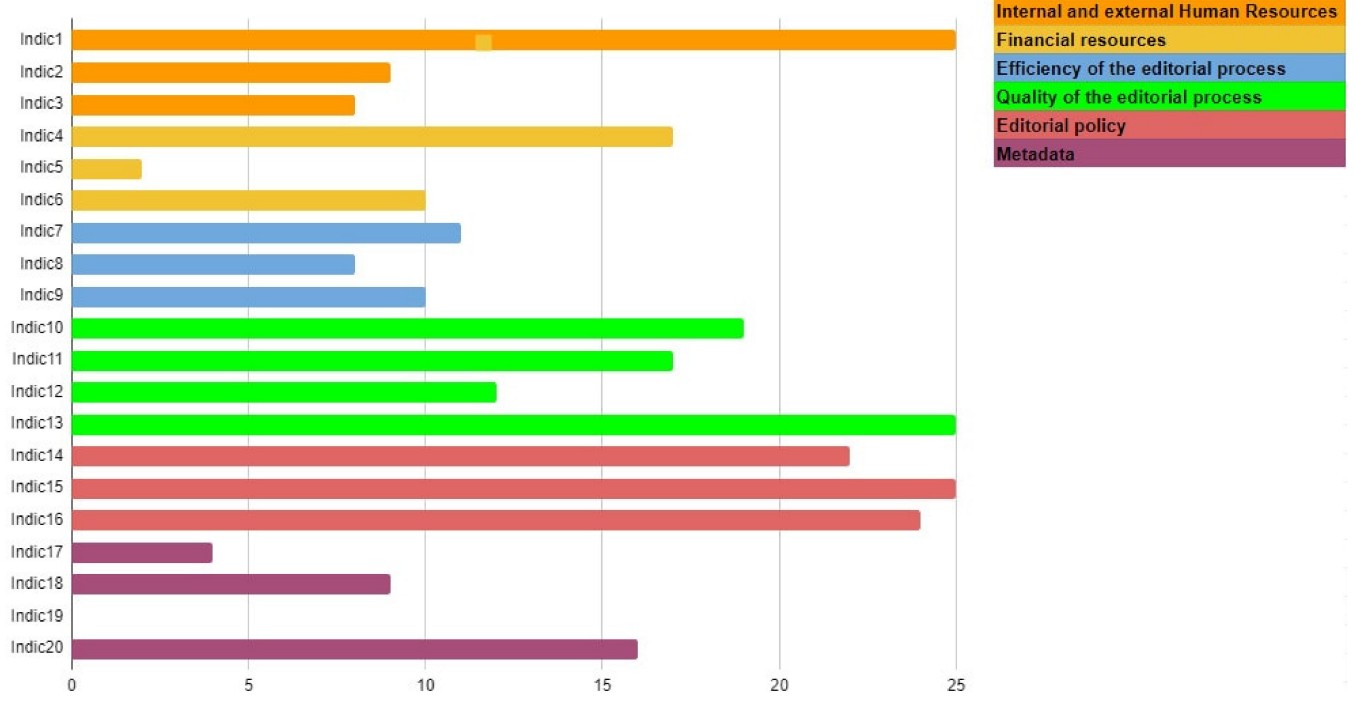

**Figure 1.** Aggregate values of the indicators.

Below is a possible interpretation of the indicators with the highest and lowest compliance. Publishing the list of reviewers does not seem to be a common practice of scholarly journals. From our point of view, for increasingly global journals and bearing in mind the debate about what a predatory journal is, the greater the transparency in the editorial process, the better. At one extreme, we might find open peer review and the debate about anonymity and its validity today [12], but at least making known which people have acted as reviewers serves to support and validate the editorial process. Thus, we would point out the need for databases and quality agencies that evaluate journals to promote such indicators to help better understand and delimit what is a good practice and what is a quality scholarly journal.

As far as cost is concerned, again, this is clearly not a common practice. We would propose that this is a valid indicator for two reasons. Scientific policy related to open science and Plan S is concerned with ensuring that the cost (and the profits) are adjusted to the market. Likewise, the costs have to be sustainable in an ecosystem based on public funding, where austerity and control of expenses can directly affect research budgets.

The need for journals to have open-data policies may be less pressing. In this sense, the type of data and the disciplines themselves may help explain the low compliance. However, in the future, it seems that most strategies, plans, and funding bodies will require the sharing of data, and journals have to be clear about their strategy in relation to this, as Palmer [29] and García-García et al. [30] also point out. The debate about who hosts and

curates datasets is especially relevant when it comes to responsibility for privacy, legal, and ethical issues related to supplementary materials. This explains, for example, how even in publications related to the COVID-19 pandemic, there have been low levels of data sharing [31]. A similar explanation would also make sense in relation to the criterion on self-citations. Even though none of the journals studied complied with this indicator, we think that it should remain on the list. For example, there was controversy recently stemming from a study developed by the National Agency for Quality Assessment and Accreditation of Spain (ANECA) in relation to the 'non-standard behaviors' of certain journals in relation to self-citation. It focused on the difficulty of setting and understanding, as often happens with plagiarism, a clear border between what is considered correct and what is not [32]. Certain databases use the amount of self-citation as a criterion. Knowing if a journal has practices that could affect whether it is included in a database or not is something that should, we believe, be known to authors before they submit their articles. It would also help improve monitoring of this behavior.

The second result we want to highlight is that our assessment system also allows for a more global analysis of the parameters. We can analyze the level of consolidation of the aspects in the corpus of journals included in the study: which indicators scored worst in this analysis, which scored best, and which obtained intermediate results. We also propose a visual representation of this analysis of compliance with the parameters through the creation of a vertical bar chart showing the value obtained for each parameter (as shown in Figure 2). This allows for a nominal comparison of the values associated with each of the parameters.

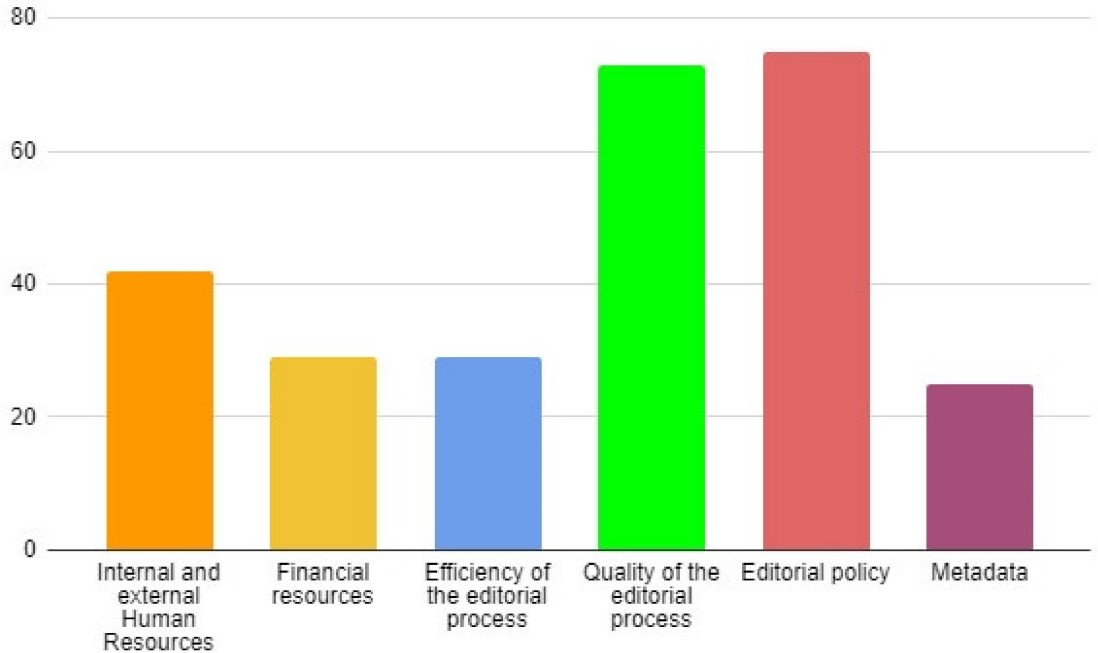

**Figure 2.** Levels of compliance with or implementation of the parameters.

If we apply this analysis to our study, we can generate the corresponding graph (Figure 2) and see the levels of compliance with or implementation of the six parameters in the selected corpus of journals. As we can see, some areas are strongly consolidated, such as 'quality of the editorial process', and 'editorial policy'. In this particular case, this may be because it is one of the quality criteria applied by DOAJ, FECYT, Web of Science, and others. However, other parameters (such as 'financial resources', 'efficiency of the editorial process', or 'metadata') have some way to go to raise the level of transparency of the analyzed journals.

The third result we want to highlight is that our analysis system also lets us generate a general transparency index for the best journals in the selected corpus. The best journals are those above the average general transparency score for the corpus. We can analyze the general level of transparency of these journals by aggregating their scores for all the indicators and generating a ranked distribution of the journals based on the quantitative value obtained. We propose a visual representation of this analysis of the general transparency index of the best journals in a vertical bar chart showing the value obtained by each journal when aggregating the total scores of the indicators. By adding a line, we can compare this index for each journal against the average general transparency value of the journals in the analyzed corpus (as seen in Figure 3). This enables a nominal comparison of the values of this index, ranking this subset of the best journals in the corpus according to the transparency index and showing how they are above the average general transparency level of the corpus.

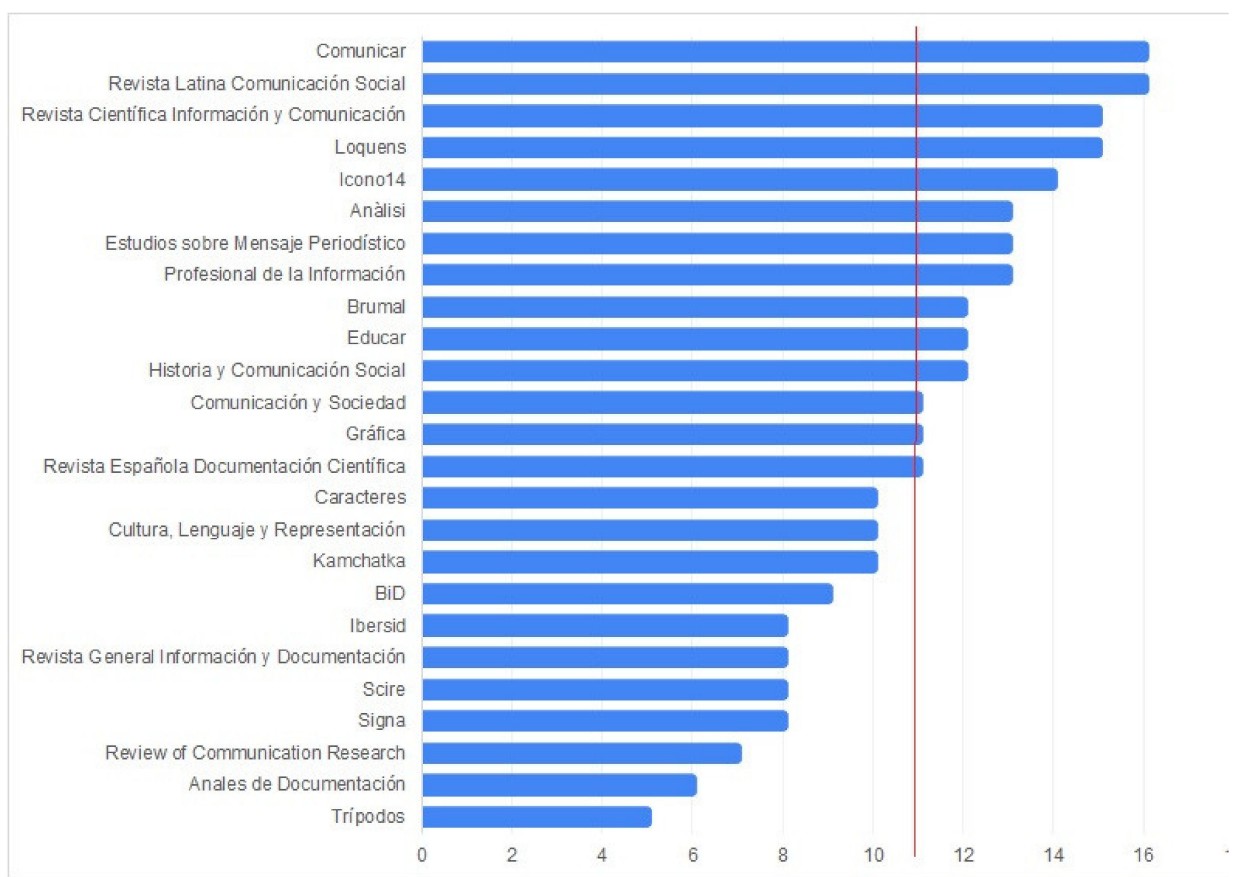

**Figure 3.** Journals' compliance with indicators (I).

If we apply this analysis to our study (25 journals from the fields of Communication and LIS), we can generate the corresponding graph (Figure 3) and compare the general transparency indices of the best journals in the selected corpus (14 in total). By introducing a line that codifies the average general transparency index of the corpus (10.92), we can see how far each journal is above this average value.

The fourth result we want to highlight complements the previous one: our analysis system also lets us generate a general transparency index for journals with the lowest compliance in the selected corpus. These journals are those below the average general transparency score for the corpus. We can analyze the general level of transparency of these journals by aggregating their scores for all the indicators and generating a ranked distribution of the journals based on the quantitative value obtained. We propose a visual representation of this analysis of the general transparency index of the worst journals in

a vertical bar chart showing the value obtained by each journal when aggregating the total scores of the indicators. By adding a line, we can compare this index for each journal against the average general transparency value of the journals in the analyzed corpus. This enables a nominal comparison of the values of this index, ranking this subset of the worst journals in the corpus according to the transparency index and showing how they are below the average general transparency level of the corpus. By introducing a line that codifies the average general transparency index of the corpus (10.92), we can see how far each journal is below this average value.

The fifth result we want to highlight is that our assessment system also lets us perform a distribution analysis of the selected journals in the corpus, using the number of indicators each publication complies with. We can analyze how these journals are distributed in four quartiles, where the first quartile has the best journals according to this index, and the fourth quartile has the worst. We propose a visual representation of this distribution analysis of the journals according to the number of indicators they comply with by creating a box-and-whisker plot showing how the journals are distributed in the resulting quartiles (as shown in Figure 4). This enables us to see whether the distribution is symmetrical, or if the journals are clumped together in the lower or upper levels of compliance area of the level of indicators complied with, or whether there are outliers.

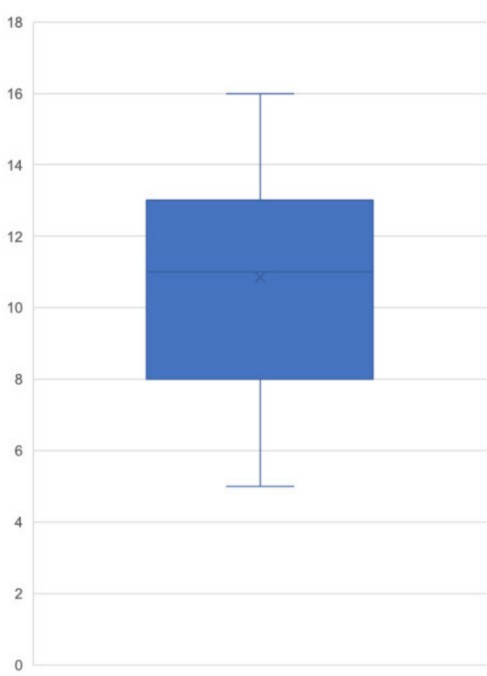

**Figure 4.** Plot of compliance with the indicators.

If we apply this analysis to our study, we can generate the corresponding graph (Figure 4) and see the distribution of the entire set of journals in terms of the number of indicators they comply with. As we can see, the diagram shows that the distribution of the journals is fairly symmetrical, with a similar proportion of journals in all four quartiles. The journals comply with a minimum of 5 indicators and a maximum of 16 indicators. The average and median compliance of the indicators is very similar, at around 11.

The last result we want to highlight is that our assessment system also lets us study the relationship between the level of transparency and other external quantitative indicators applicable to the journals in the selected corpus. We can see whether such correlation exists, and if so, whether the correlation is positive or negative. Thus, we observe a certain degree of visual correlation from the scatter plot and the trend line included in this graph. We propose a visual representation of this analysis in the form of a scatter plot showing each journal as a point placed along the X and Y axes based on the numerical values of

the journal's transparency index and the quantitative value of the other external indicator selected (as shown in Figure 5). This graph can be completed by adding a trend line to highlight the correlation.

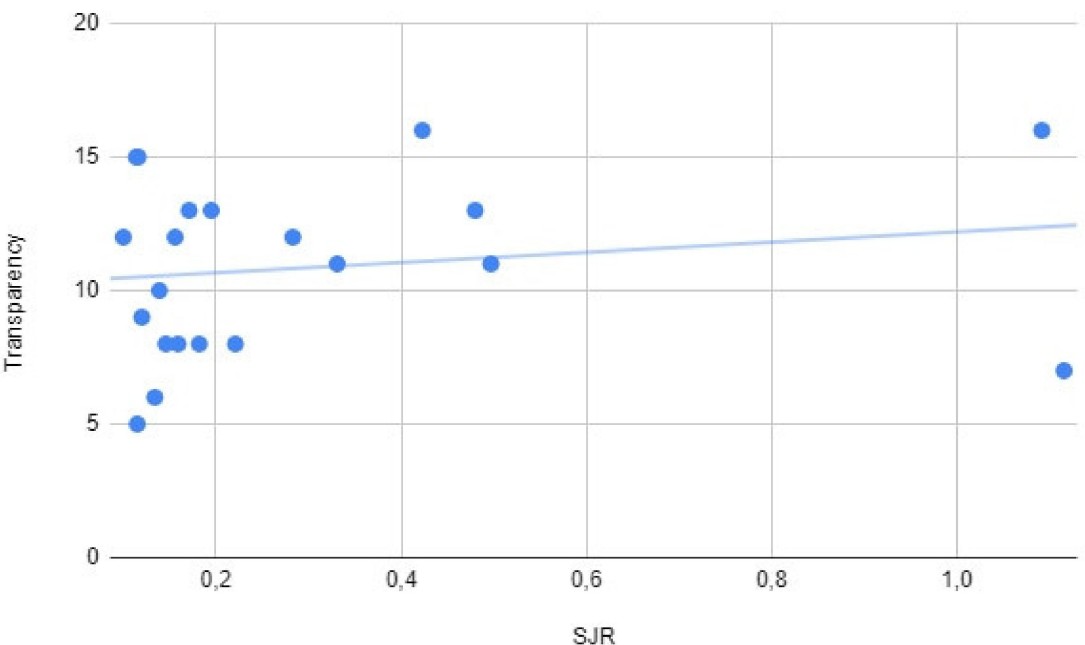

**Figure 5.** Diagram of the correlation between the compliance value and the Scimago Journal & Country Rank.

If we apply this analysis to our study, we can generate the corresponding graph (Figure 5) and see the correlation between the transparency index of the journals in the corpus and their impact factor (specifically, the Scimago Journal & Country Rank (SJR)). As we can see in this graph, the trend line shows that there is a certain degree of positive correlation (although not a very strong one) between a journal's SJR and its level of transparency.

## 4. Conclusions and Recommendations

Our main conclusions are as follows. First, regarding the choice of indicators:

-   The distribution of results confirms that the choice of criteria seems appropriate: the values do not all show high compliance or low compliance. There are different results that point to the possibility of comparing journals: we can see a progression and where improvements are needed.

Second, with regard to the chosen sample, but with a possible correlation for the validity of transparency analysis:

-   The indicators relating to editorial policy and to journal quality are the ones with the highest levels of compliance. This relates to the requirements for secondary databases and enables us to identify a set of quality journals in different disciplines. In this sense, then, it can be pointed out that there is a consensus for the quality indicators that a journal must meet, and the great competitiveness between journals validates compliance internally (sustainability of the journal in relation to the funders) and externally (placing in quartiles of the databases)
-   The indicators relating to metadata present a clearer area for improvement. This can also be explained by the fact that certain criteria are recommended but not required by journal indexers. Likewise, there is a need for clear metadata policies that allow for the interoperability of scholarly articles and the ability to apply data-mining and knowledge-extraction mechanisms that are only possible with quality metadata. Plan S, for example, seeks to achieve quality standards, although, for the moment, it

has placed them in the field of non-mandatory supplementary indicators for journals that must or want to comply with Plan S [22].

- Indicator 19 (monitoring self-citation) at 0 and indicator 5 (itemizing costs of the publication) at 1 point are the lowest on the analysis (Figure 1). There is full compliance with indicators 1 (editorial board), 13 (indexing), and 15 (license type). There is a wide range of indicators in an intermediate position.

Finally, in relation to the specific sample chosen for study:

- The *Revista Latina de Comunicación Social* and *Comunicar* are clearly ahead with over 75% compliance with the indicators studied. At the other extreme, *Anales de Documentación* and *Tripodos* are the journals with the lowest level of compliance.
- There is significant room for improvement for journals to openly provide the information they may already have and the criteria they apply. In other cases, they could consider including them. Indeed, we believe that the indicators could be used by journals as a self-assessment tool for ongoing improvement.

The possible limitations of the study come from the choice of the sample and the discipline. The sample should be expanded, and rankings comparing academic journals could be created. The journals themselves could include, as part of their best practices, icons showing their compliance with the indicators/parameters. This would make it easier to quickly see their compliance with Plan S, transparency, etc., without having to comb through large amounts of information published in widely differing ways. The proposed indicators allow for analysis and verification of the transparency of academic journals, and they can help interpret the transparency of these academic journals (many of which are from academic publishers receiving public funding).

**Author Contributions:** Conceptualization, A.L.-B., M.V., and C.O.; methodology, formal analysis, investigation, data curation, writing—original draft preparation, A.L.-B., M.V., C.O., and M.P.-M. All authors have read and agreed to the published version of the manuscript.

**Funding:** This research and the APC was funded by Ministerio de Ciencia, Innovación y Universidades, grant number RTI2018-094360-B-I00.

**Data Availability Statement:** The data presented in this study are available in https://doi.org/10.6084/m9.figshare.17060336.v1.

**Conflicts of Interest:** The authors declare no conflict of interest.

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
