# Peer review of "Publisher Transparency among Communications and Library and Information Science Journals: Analysis and Recommendations"

_publications, doi:10.3390/publications9040054_

Round 1

Reviewer 1 Report

The introduction tells little specifics about why the model presented by the authors is needed and how the an indicator can increase the transparency of a journal. The number of scientific sources used here also seems small compared to the significance of what is being said. In the Methodology section, the authors only describe the indicators used, but do not show why these are the most suitable for exploring the value of a journal, or whether any order should be formed or weights assigned to these. Figure 1 and the explanation of the results in general are weak. In the Results section, the text has little information content, the authors do not explain the results thoroughly. What does Figure 1 show? What does this add to Table 1? The additional figures cannot be interpreted in the absence of appropriate markings or are too small (e.g. Figure 3). Nor is it clear exactly how the authors obtained these figures, a description of which is also missing.

Author Response

Various changes have been made to the introduction to clarify this issue, thank you very much. We count on having known how to incorporate them correctly.

In the design of the research, the possibility of weighting each of the indicators was a question posed. It was considered more appropriate to make a list where all the indicators had the same weight as a measure of the fact that it was the total consideration of the indicators and not the total sum that made sense for each journal. Moreover, We do not want to specify a number or delimit a limit but to give a checklist to improve, as Plan S does. Parameters as well as indicators do not make sense on an individual basis, as value is the sum of them in order to improve transparency. We have added information in the methodology to clarify this issue. Thanks for the comment.

Figure 3 has been changed and the purpose of Figure 1 has been better explained. As for how the results have been obtained, these are described in the methodology. The results have also been made available on the Internet for consultation.

Reviewer 2 Report

Overall

The set-up of the study is interesting, but there are serious issues with the English, lack of citations and evidence to back up claims, lack of discussion of the results, failure to follow through on all goals, and lack of explanation about the methodology.

Examples of issues with English and style: 

Line 1:

The primary goal of the research study is to analyze

Line 34 communication

Frequent improper use of the word “thus”

Furthermore, academic journals also have a wide range of internal data which is often invisible, such 45 as their budgets, the number of articles rejected each year, the number of reviewers per article, average response time after article submission, and so 46 on.

Line 79 unclear antecedent (this)

Such indicators would not relate to the content of the scientific articles or 79 the supplementary data, but to the information, the journal itself supplies on the publication process.

Repeated use of the word “undoubtedly” to signal a standard view, when the information should really be better explained or backed up with evidence.

Title

The title says “scientific journals.” I suggest making that align more with what was studied- Library Science and communications journals.

Abstract

It is not initially clear, in what practices or information you are seeking “transparency.” Transparency in editorial leadership and peer review? In OA practices? In selection? An additional sentence, or brief explanation of what type of information should be presented would be helpful. 

Introduction

Line 26-28- how is open access a “value”? OA is a practice that may embody or be based on some values but it is not a “value” in itself.

Line 29-31- Who is calling in to question the validity of JIF? Can you present research to back this up?

1.1

There is a lot of information here that is given very quickly without much explanation, context, or development.

There are a lot of terms given here that are not defined or contextualized- open access, open science, open data, open government, open data environments. These are potentially related terms, but with distinct meanings.

How does OA change scientific policies and business models? Is this in reference to publishing?

The reference to transparency in management in the public sphere seems out of place and antidotal. 

Line 58-59, why is it important for PhD. Students?

Line 60- what are the models?

Line 64-evidence for these factors being critical?

Line 65- what agencies are changing and how?

Line 69, which indicators are common?

Line 71, what do you mean “managing public resources”? Is this a reference to publicly funded research and what that money is spent on?

1.2

Line 81-82- what does this mean? Are you referring to things like Scopus or Scimago?

Lines 83-84- it would be helpful to further explain here how your research is different than what is available from businesses like Scopus or Cabell’s and why that is needed in an OA format.

Line 87

As a preliminary measure to test the effectiveness and capacity of such a classification checklist to differentiate between transparent and non-transparent journals in an ecosystem of similar journals, we propose analyzing the academic Information and Communication journals indexed in Scopus in order to see whether these journals disclose the management and editorial data they generate.

Throughout- you use several names to describe the category of journals analyzed. “Information and Communication,” “Communication and Documentation,” “Communication and Library and Information Science” etc. I suggest using the exact wording from Scopus or Scimago, or at least the same wording every time.

Line 95- Reordering these goals might help

  1. To create a closed set of indicators for studying and comparing the transparency of scientific journals.
  2. Apply these indicators to Spanish Communication and Documentation journals indexed in Scopus.
  3. Use the knowledge gained from that evaluation to develop a proposal to improve journals, enabling them to have transparency policies.

Why did you use Scimago to identify journals, and Scopus to review them?

Methodology

Did the evaluators perform their analysis blind to each other?

Spell out the acronyms DOAJ, FECYT, WOS.

Beyond The 20 indicators are very vague, I would like to see a greater explanation of what the evaluators were looking for.  Many of these are not self-evident

It is unclear where evaluators were looking for the information. Where are they looking on DOAJ, FECYT, WOS, SCOPUS, Plan S.? Or, were they looking on the journal’s website? If not the journal site- why not? If you are only looking at external platforms, are you not just evaluating what those platforms gather, rather than what the journals make available?

Table 1- It is unclear what this table is for- Which data points are listed on each platform?

Results and Discussion

125-128- the sentence is unclear

Unclear how you analyzed the level of consolidation- is this an average of the indicators in that area? A standard deviation? Some other statistical measure?

Figure 3- the title should indicate why there is only a portion of the sample here. For example “Journals with transparency scores at or above the average” same for table for “Journals with transparency scores below the average. Additionally, the names of the journals are too small to read, and a different chart design would be better.

The presentation of each result is somewhat wordy and confusing. It is not typical to say “Also relating to this fourth result, we propose a visual 192 representation of this analysis…”

For figure 6- did you perform a statistical measure to test the correlation such as a Chi-Square test? If not, then you should not use language implying a positive or negative correlation or the significance of such.  Why does this graph show a positive degree of correlation?

Missing a discussion of the results. The sentences describing the results are unclear and too long and do not give much discussion about what these results mean, or how they relate to previous studies.

Conclusions and recommendations

#1- criteria selection- this seems antidotal and needs a clearer explanation of why they are appropriate. If two of the indicators have 0 and 1 journal reporting them- why are they still valid indicators?

#2- this information is interesting, but could use some re-wording to make it more clear.

#3- a little more discussion about the journals and how it relates to their SJR etc. would have been interesting.

These outcomes do not connect back to your first goal of the paper “To develop a proposal to improve journals, enabling them to have a transparency policy.”

Author Response

Thank you very much for your comments, we think you have made this new version much better than the previous one.

In the attached file you can find the answers and the changes made thanks to your work

Reviewer 3 Report

Dear Authors,

I enjoyed reading your paper on this interesting topic.  As to suggestions for improving it, I suggest expanding a few sections and a reconsideration of your title to be more aligned with the paper's content. 

It could be helpful to readers outside Europe to explain Plan S and its impact on scholarly communication. Additional elaboration on metrics and perceptions of impact affect not only your fields but also those in high stake or time sensitive research areas. 

Readers may not be familiar with FECYT and possibly the other platforms mentioned in the paper. 

Could you also reveal more as to why the 25 journals were selected for your "corpus"?  This would be important for anyone wishing to replicate your study, or use it as a model for journals in other disciplines. 

Importantly, I believe there ought to be descriptions of each of your Indicators, and so eliminate ambiguity. 

As to possible ramifications of your study, you might look at library products such as Ulrichsweb Global Serials Directory or Cabell's Directory of Publishing Opportunities as a point of comparison for representation of metrics, acceptance rates and so on. 

Also from the library and collection management angle, your framework for analyzing journal quality vis-a-vis Indicators could be applied to the selection or de-selection process in the event of additional budget losses. 

I mention a possible title revision because some readers will equate "scientific journals" with STEM journals rather than generalized research and knowledge construction.  So a title such as  "Publisher transparency among Communications and Library and Information Science journals: analysis and recommendations" might be a better fit or possibly a matter to discuss among yourselves.  I see merit in this paper and hope to see it in publication after some improvements are made.

Author Response

(The authors gave the same response as above.)

Reviewer 4 Report

This article argues for transparency of editorial information as an aid to potential authors in selecting journals for submission. While there may be some value in this I feel that the article as presented has a number of shortcomings.

In the title and throughout the article the authors use the term “scientific” (scientific journal, scientific communication, scientists, etc.). In English, unless otherwise qualified, this would normally imply the hard sciences, whereas in this case the discipline concerned is library studies. It would seem that the word “academic” would be more appropriate, which is indeed the term the authors use in their keywords.

The study is restricted to Spain. It is not clear why Spain should have been chosen (other than the fact that the authors all work at a Spanish university). Is there any reason why Spain should be considered typical of academic journal publication in other countries? We need some justification for the restriction of the study to this single country.

The article is also restricted to the field of Communication, Library and Information studies. Why has this field been chosen? Can the results for these journals be taken as indicative of academic journals as a whole? Moreover, do these journals publish in Spanish or in English (or in other languages)? Given that over 90% of academic journal publication in the hard sciences is in English, how can the choice of this field in Spanish-based journals be justified?

Can all the indicators be given the same weight? There can be genuine reasons (related to confidentiality) for not revealing information about reviewers, for example.

There are a few minor errors in the English; e.g. §3 “which indicators” should be “whose indicators”.

In general, I feel that the article was not sufficiently explicit, and frequently difficult to follow. The authors need to spell out their argument more precisely and clearly.

If the authors feel that they can overcome the difficulties outlined in this report, they may wish to revise and resubmit.

Author Response

(The authors gave the same response as above.)

Round 2

Reviewer 1 Report

Thank you for your revision.

Author Response

Thank you very much for your review.

The authors

Reviewer 2 Report

I am not satisfied that the writers made a serious effort to address the concerns given. Additionally, I was confused that several times the writers explained their choices in the letter, but did not add the same content to the manuscript. I've attached my responses and further comments in red text.

Author Response

(The authors gave the same response as above.)

Reviewer 4 Report

The authors have done a great deal to address the reservations I expressed in my first report, and large sections of the paper have undergone extensive rewriting. Although the paper is a little repetetive at times, I feel that this version is of sufficient interest to merit publication.

I would recommend the following minor modifications:

l.31 open > Open

l.39 commu-nication > communication

l.191 which journals > which ones journals

l.279 or if > or whether

Author Response

Thank you very much for your review. Your comments have been considered

The authors

Round 3

Reviewer 2 Report

The authors have put in a more sincere effort in this third draft, and there is a significant improvement. I am impressed by the addition of relevant literature and improved clarity on the research design and results. However, there are still significant issues with the English writing and clarify of much of the manuscript. 

The authors may consider reviewing  Adrian Wallwork’s English for Writing Research Papers (2016) or other helpful resources like http://www.ref-n-write.com/trial/research-paper-sample-writing-introduction-section-academic-phrasebank-vocabulary/. Even as a native English speaker, I find these types of resources extremely helpful in my own work to write clearly and succinctly.

Author Response

Thank you very much for your comments, we believe that the new version solves the aspects related to the language